# Regional Differences in Uptake of Vaccination against COVID-19 and Influenza in Germany: Results from the DigiHero Cohort

**DOI:** 10.3390/vaccines11111640

**Published:** 2023-10-26

**Authors:** Bianca Klee, Sophie Diexer, Myka Harun Sarajan, Nadine Glaser, Mascha Binder, Thomas Frese, Matthias Girndt, Daniel Sedding, Jessica I. Hoell, Irene Moor, Michael Gekle, Rafael Mikolajczyk, Cornelia Gottschick

**Affiliations:** 1Institute for Medical Epidemiology, Biometry and Informatics (IMEBI), Interdisciplinary Centre for Health Sciences, Medical Faculty of the Martin Luther University Halle-Wittenberg, Magdeburger Str. 8, 06112 Halle (Saale), Germany; bianca.klee@uk-halle.de (B.K.); sophie.diexer@uk-halle.de (S.D.); myka.sarajan@uk-halle.de (M.H.S.); nadine.glaser@uk-halle.de (N.G.);; 2Department of Internal Medicine IV, Oncology/Haematology, Martin Luther University Halle-Wittenberg, Ernst-Grube-Str. 40, 06120 Halle (Saale), Germany; mascha.binder@uk-halle.de; 3Institute of General Practice and Family Medicine, Interdisciplinary Centre for Health Sciences, Medical Faculty of the Martin Luther University Halle-Wittenberg, Magdeburger Str. 8, 06112 Halle (Saale), Germany; thomas.frese@uk-halle.de; 4Department of Internal Medicine II, Martin Luther University Halle-Wittenberg, Ernst-Grube-Str. 40, 06120 Halle (Saale), Germany; matthias.girndt@uk-halle.de; 5Mid-German Heart Centre, Department of Cardiology and Intensive Care Medicine, University Hospital, Martin Luther University Halle-Wittenberg, Ernst-Grube-Str. 40, 06120 Halle (Saale), Germany; daniel.sedding@uk-halle.de; 6Paediatric Haematology and Oncology, Martin Luther University Halle-Wittenberg, Ernst-Grube-Str. 40, 06120 Halle (Saale), Germany; jessica.hoell@uk-halle.de; 7Institute for Medical Sociology, Martin Luther University Halle-Wittenberg, Magdeburger Str. 8, 06112 Halle (Saale), Germany; irene.moor@uk-halle.de; 8Julius-Bernstein-Institute of Physiology, Medical Faculty of the Martin Luther University Halle-Wittenberg, Magdeburger Str. 6, 06110 Halle (Saale), Germany; michael.gekle@uk-halle.de

**Keywords:** COVID-19, influenza, vaccine hesitancy

## Abstract

During the COVID-19 pandemic in Germany, vaccination uptake exhibited considerable regional disparities. To assess the factors contributing to this variation, we examined the association of sociodemographic variables with COVID-19, COVID-19 booster, and influenza vaccination status within a cohort of 37,078 participants from 13 German federal states in the digital health cohort study commonly known as DigiHero. Our findings revealed variations in vaccination rates based on sociodemographic factors. However, these factors had limited explanatory power regarding regional differences in vaccine uptake. In contrast, we found substantial correlations between regional support of specific parties during the last local elections and the vaccination uptake at the level of each administrative district. In conclusion, sociodemographic factors alone did not suffice to explain the regional disparities in vaccine uptake. Political stances can play a major role, although the current investigation did not assess individual political orientations but rather used only an ecological approach.

## 1. Introduction

Vaccination against SARS-CoV-2 became available in the European Union at the end of 2020. In the summer of 2021, most of the German population had received basic immunization. Subsequently, the first booster vaccination became available towards the end of 2021. The spread of the Omicron variant was observed in numerous countries across the globe, including Germany, where the Robert Koch Institute (RKI)—the federal governmental agency in Germany tasked with the control and prevention of diseases—documented 16,748 Omicron cases as of 30 December 2021 [1]. Thus, the RKI advised individuals who had not yet received their first COVID-19 vaccine doses to become vaccinated and recommended that booster doses should be accessible to all age groups [1]. The second booster vaccination, optimized for Omicron, became available in the autumn of 2022.

The introduction of available vaccinations against COVID-19 in 2021 met with strong resistance from some parts of the German population [2]. This is known as vaccine hesitancy and it is characterized as “the delay in acceptance or refusal of vaccination despite availability of vaccination services” [3]. Vaccine hesitancy is not a novel phenomenon and it did not emerge during the COVID-19 pandemic. It has been observed in the majority of countries and differs across country income level [4]. Hesitancy towards vaccinations is fueled by fears regarding vaccine adverse events or side effects [5]. At the same time, it is crucial to differentiate vaccine hesitancy from the risk-benefit assessment of vaccines. Vaccine hesitancy focuses on people’s attitudes and perceptions regarding vaccination [3], while the risk-benefit assessment involves a systematic comparison between the risks and benefits of vaccination [6]. This is essential for informing public health decision-making [6]. Both aspects of hesitancy and risk-benefit assessment can play a role in vaccine uptake. An additional aspect of judging risk-benefit is the effect of vaccinations on the spread of infection, thus going beyond individual perspective. Other than the COVID-19 vaccine, some vaccines that face opposition among the public include the Measles, Mumps, and Rubella (MMR), HPV (Human Papillomavirus), Hepatitis B, Influenza, and Polio vaccines [7,8]. Nevertheless, opposition towards COVID-19 became part of a political ideology in some countries [9].

In Germany, there are notable disparities between former Eastern and Western federal states with respect to vaccinations. Vaccine hesitancy towards COVID-19 vaccines had a particularly noticeable presence in the former Eastern German states [2]. Eastern German states tend to have stronger support for right-wing parties, while Western German states lean more towards centrist political parties [2,10]. AfD (Alternative for Germany), the right-wing German political party, along with other protest parties that emerged during the COVID-19 pandemic, have made public their resistance to government measures and the associated vaccines [2]. As of April 2023, the overall vaccination coverage in Germany was 77.9%, with people of all ages receiving at least one dose [11]. Furthermore, 76.4% of the total population received basic immunization, 62.6% received one or more booster doses, and 15.2% received at least two booster doses [11].

Due to respective recommendations from the STIKO (Standing Committee on Vaccination), a higher proportion of those aged 60 and above receive influenza or pneumococci vaccinations [12]. During the winter of 2021/22, 43.3% of individuals aged above 60 and 35.4% of individuals aged above 18 (with relevant underlying conditions) were vaccinated against influenza [12], with a higher vaccination coverage rate against influenza observed in Eastern Germany than in Western Germany [13]. For other vaccinations such as Pertussis/Diphtheria/Tetanus, the vaccination coverage rates were also considerably higher in former Eastern German states than in former Western German states [14,15]. This outcome is probably attributed to distinct attitudes associated with these established immunizations [14,15].

The outcome of vaccination uptake can be driven by both risk-benefit assessment and vaccine hesitancy. We hypothesized that regional differences in vaccination uptake are rather an indicator of vaccine hesitancy, while differences across age groups are rather an example of risk-benefit considerations. For other individual factors, like education, the distinction between both of these categories is less clear. Furthermore, theoretically, even knowledge of regionally different behavior can affect risk-benefit estimates, so hesitancy and risk-benefit considerations remain intertwined, and regional differences are only an indirect indicator of hesitancy. Given this contextual background, our objective was to investigate how much of the regional differences across federal states or within regions in Germany could be explained by differences in sociodemographic factors for COVID-19, COVID-19 booster, and influenza vaccinations. Furthermore, we studied whether regional differences in vaccine uptake were associated with support for specific political parties at the regional level. 

## 2. Materials and Methods

### 2.1. Study Design and Questionnaires

The data used in this study were collected in the population-based prospective cohort study for digital health research in Germany (DigiHero, DRKS Registration-ID: DRKS00025600) [16]. DigiHero started in the city of Halle (Saxony-Anhalt, Germany) in January 2021 and was later extended to other federal states in Germany, with a strong focus on Middle Germany. Random samples were derived from registration registries and the selected individuals were invited through postal letters to DigiHero. Participation in this study was online and participants were invited to complete various questionnaires three to four times per year. In the first questionnaire, sociodemographic information was collected. The Ethics Committee of the Martin Luther University Halle-Wittenberg (2020-076) approved this study and informed consent was obtained from all participants during online registration.

In the spring of 2023, we invited 70,538 participants from 13 federal states in Germany who had been recruited until then to answer questions regarding vaccinations against COVID-19 and influenza in the preceding autumn/winter season. Additionally, we collected information on all vaccinations against SARS-CoV-2 since the beginning of the pandemic. In our study, individuals who have received four doses or more are considered to have received a SARS-CoV-2 booster vaccination. We combined these data with sociodemographic information from the initial questionnaire. We categorized education levels based on the International Standard Classification of Education (ISCED-97) into three categories: low, medium, and high [17]. In the baseline questionnaire, we inquired about net household income, providing seven income categories. To determine net household equivalent income, we first computed the mean of each category and divided the mean by the sum of the household weights. Household weights were calculated using the following weights: the first adult was assigned a weight of “1”, while all other adults (individuals aged 14 and above) held a weight of “0.5”, and children were assigned a weight of “0.3”. The resulting value was then categorized using the initial seven income categories.

We then retrieved results from the state elections in Saxony in 2019 and Saxony-Anhalt in 2021 at the administrative district (Landkreis) level from official election sources [18,19]. 

### 2.2. Statistical Analysis

Descriptive statistics for this study are presented using frequencies and percentages. We employed generalized additive models (GAM) to effectively model vaccination uptake by age. We examined three outcomes: (i) receiving any vaccinations against SARS-CoV-2 versus none; (ii) receiving any booster vaccinations against SARS-CoV-2 versus none; and (iii) receiving an influenza vaccination in autumn/winter 2022/23 versus not receiving one. We examined the vaccination uptake across federal states initially without adjustments and subsequently accounting for sociodemographic factors such as age, sex, place of residence, urban/rural place of living, migration background, net household equivalent income, and education by a stepwise approach. We used the order of importance of the adjustment factors determined for any vaccination for the other outcomes as well. 

For the ecological analysis, we examined the regional variations in vaccination uptake within the federal states of Saxony and Saxony-Anhalt, both of which had participants from all administrative district units (Landkreis) in the DigiHero study. We then correlated these variations with the percentage of support for political parties.

## 3. Results

Out of the individuals invited for this study, 38,827 (53.7%) completed the vaccination questionnaire between March and April 2023. We removed participants with incomplete vaccination data (*n* = 1749), resulting in a final sample of 37,078. Those who responded to this questionnaire showed some differences in terms of sociodemographic data compared to those who did not participate, mainly a lower response rate among males, individuals in the youngest age group, and participants with low education and low income (Table 1).

Differences in the uptake of SARS-CoV-2 and influenza vaccinations were observed across various sociodemographic characteristics and regions (Table 2). The overall vaccination uptake was highest for any COVID-19 vaccination (92.8%), followed by influenza vaccine (44.1%), and booster vaccination for COVID-19 (25.6%).

The association between age and vaccination uptake displayed different patterns for the three studied vaccines (Figure 1). It is interesting to note the lower uptake in the age group of 30 to 60 for all vaccinations. Another observation is the low booster uptake below 60 years, and a nearly linear increase in uptake with age for influenza vaccination. 

Vaccination uptake for SARS-CoV-2 was lowest for Saxony, followed by further federal states in the former East Germany (Table 2). In Schleswig-Holstein, the uptake of booster vaccination was much higher than in the other federal states. For influenza vaccination, the uptake was highest in the city of Hamburg, as well as the federal states of Saxony-Anhalt and Schleswig-Holstein. 

In general, the differences in sociodemographic variables did not explain much of the regional variation in vaccine uptake (Figure 2). However, when city residence was included in the models for COVID-19 vaccination, the estimates for Berlin, Hamburg, and Stuttgart changed substantially. In contrast to the COVID-19 vaccines, influenza vaccination uptake was, among the studied federal states, highest in Saxony-Anhalt. 

Within the federal states of Saxony and Saxony-Anhalt, similar regional patterns emerged for uptake of COVID-19 and influenza vaccines (Figure 3). Across the seven strongest parties, for three existed positive correlations (Figure 4, with substantial fraction of variation explained, Figure 5), for three rather minor, and for one party a strong negative correlation. The latter is a right-wing party which holds strong anti-vaccination policies.

## 4. Discussion

Our study aimed to investigate whether differences in sociodemographic characteristics could account for regional variations in uptake for COVID-19, COVID-19 booster, and influenza vaccines. National reporting provides data on vaccination rates based on age, but it does not include factors such as education and income. Despite considering these sociodemographic variables, the substantial differences in vaccination uptake across different federal states remained largely unchanged. The regional uptake showed a rather strong correlation with the regional proportion of first votes in the last local elections, with three parties with a positive correlation, two with no correlation, and one with a strong negative correlation in the two federal states studied. These correlations explained a considerable fraction of the variation.

Sociodemographic factors did not explain much of the regional variations. In fact, it appears that beliefs related to vaccination are more profound and related to personal preferences. An in-depth qualitative survey involving 33 German participants identified four primary themes that influenced the decision to receive a COVID-19 vaccine: (i) assessment of the benefits and risks of being vaccinated against COVID-19; (ii) influence of existing social and political conditions; (iii) emotional responses to the pandemic and its social and political impact; and (iv) trust and confidence in health authorities and the vaccines themselves [20]. Recurrent themes across multiple studies in both Germany and Europe as a whole included reasons such as low levels of trust in healthcare authorities and COVID-19 vaccines, concerns about potential side effects, and alignment with right-wing political stances [21,22,23,24,25,26]. Thus, it appears that there are fundamental concerns about vaccination that are also intertwined with political ideologies. It is unclear in which direction the influence between politics and vaccination concerns occurs. 

Several other vaccines, apart from the SARS-CoV-2 vaccine, have faced resistance from the public. For example, the Hepatitis B vaccine and the HPV (Human Papillomavirus) vaccine are two modern vaccines that have encountered substantial resistance from the public [8]. Many of these concerns were attributed to fears regarding claims of association with autoimmune diseases such as multiple sclerosis for the Hepatitis B Vaccine and the apprehension of promoting sexual activity for the HPV vaccine [8].

Hesitancy towards the influenza vaccine was prominent even before the COVID-19 pandemic, and this hesitancy persisted when the initial COVID-19 vaccination rollout began in 2021. Less than half of the study participants were vaccinated for influenza, with the highest uptake observed in Eastern German states. This trend is consistent with previous research on influenza vaccination, which has shown higher coverage rates in Eastern Germany compared to Western Germany [13]. Vaccine hesitancy towards influenza was also seen in other countries before and during the COVID-19 pandemic. For instance, low coverage rates were observed in the United States in 2018, where 36.8% of 4286 participants were reluctant to become vaccinated against influenza [27]. Two major factors, namely race or ethnicity and gender were found to be the predicting factors for vaccine refusal [27].

The low uptake of SARS-CoV-2 booster vaccination in the autumn/winter of 2022/23 suggests that a significant portion of the German population will have received their last vaccination more than 12 months ago by the autumn/winter of 2023/24. This is concerning because, during the winter season of 2022/23, a significant number of people continued to become infected with SARS-CoV-2 [28]. While for influenza, the autumn vaccination campaign is sufficient to avoid a large burden of severe cases, influenza has a much lower basic reproduction number than SARS-CoV-2 and the immune protection for influenza was built over many years in the population. Since the protection against infection is only partial for the Omicron variant [29,30,31,32], its circulation hardly can be avoided and questions concerning the proportion of severe cases will become relevant for the health care system.

Our study had several limitations. Firstly, our data included only information on vaccination uptake, so we did not assess vaccine hesitancy. Therefore, we can only indirectly infer concerning vaccination hesitancy from regional variation, after controlling for regional variation in sociodemographic factors. Furthermore, at the individual level, vaccine hesitancy can be intertwined with an individual risk-benefit assessment. An even more complex issue results from transmissibility of infections and the risk-benefit assessment can include considerations beyond one’s own health. Furthermore, at a more technical level, we considered only the number of vaccinations for COVID-19, while particularly in the early stages of the pandemic, infections were considered to provide protection for a certain period. This approach might have led to an underestimation of willingness for vaccination. Conversely, it is possible that those who participated in this study had a more positive attitude towards vaccination, which could have resulted in higher uptake estimates in our data. In fact, when we compared our findings to national data, we found a higher vaccination uptake, suggesting that our study participants may not fully represent the general population. Additionally, relying on study participants who may be more similar to each other across federal states than the general population of these states could have led to an underestimation of differences across federal states. Moreover, our questionnaire exclusively examined sociodemographic characteristics and did not explore political viewpoints or reasons for vaccine hesitancy. In our analysis, we compared vaccination proportions among study participants with external voting data. It is important to note that this approach may be susceptible to ecological fallacy. This was a simple correlation intended to provide an overall perspective rather than draw specific conclusions about the individual study participants’ political preferences. When correlating individual data with the percentage of support for political parties in state elections, we cannot infer the political preferences of individual participants. On the positive side, our study included a large sample size and provided regional-level data. We systematically recruited participants using a population-based approach, which differs from typical web-based studies.

## 5. Conclusions

In summary, our analysis indicates that regional differences in the uptake of the studied vaccinations in Germany cannot be attributed to differences in sociodemographic factors alone. Instead, we observed different regional patterns after adjustment for sociodemographic data, and in the two federal states for which we had comprehensive regional coverage data. We also observed correlations between support of political parties and regional vaccination uptake. It appears that political stances and individual choices are rather intertwined, but this study used only ecological analysis for the assessment of this relation and could not infer individual participants’ political preferences.

## Figures and Tables

**Figure 1 vaccines-11-01640-f001:**
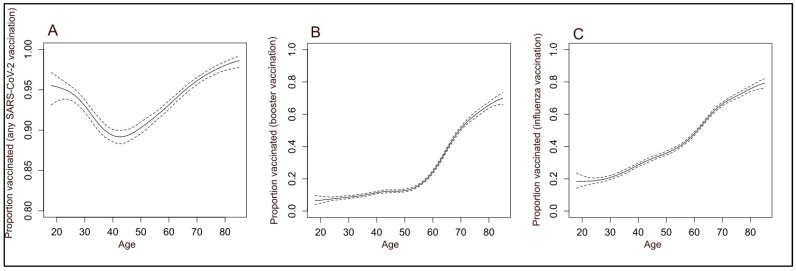
Vaccine uptake by age: (**A**) any SARS-CoV-2 Vaccination; (**B**) any SARS-CoV-2 Booster Vaccination; and (**C**) Influenza Vaccination; (splines model with continuous age; dashed lines present the 95% confidence interval).

**Figure 2 vaccines-11-01640-f002:**
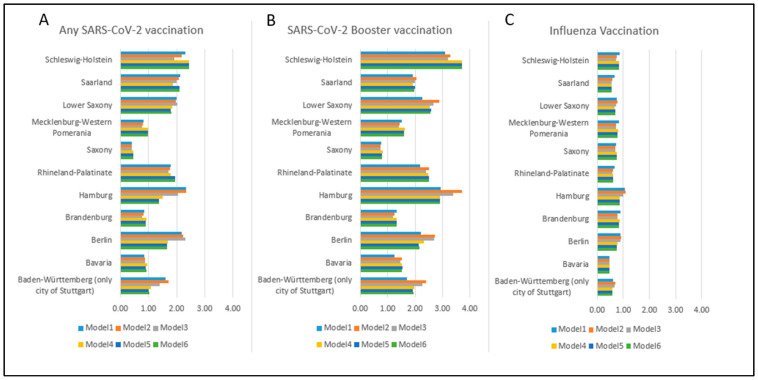
Effects of adjusting for sociodemographic variables on regional variation in vaccine uptake across different federal states in Germany: (**A**) any SARS-CoV-2 Vaccination; (**B**) any SARS-CoV-2 Booster Vaccination; and (**C**) Influenza Vaccination. The odds ratios are presented from logistic regression using Saxony-Anhalt as the reference category. The models progress from Model 1 to 6, each adjusting for an increasing number of variables: (i) Model 1 adjusts for only the federal state; (ii) Model 2 adjusts for the federal state and age; (iii) Model 3 adjusts for the federal state, age, and net equivalent household income; (iv) Model 4 adjusts for the federal state, age, net equivalent household income, and living in a city (>100,000 inhabitants); (v) Model 5 adjusts for the federal state, age, net equivalent household income, living in a city (>100,000 inhabitants), and education; (vi) Model 6 adjusts for the federal state, age, net equivalent household income, living in a city (>100,000 inhabitants), education, and sex. Additionally, the age variable was adjusted with splines.

**Figure 3 vaccines-11-01640-f003:**
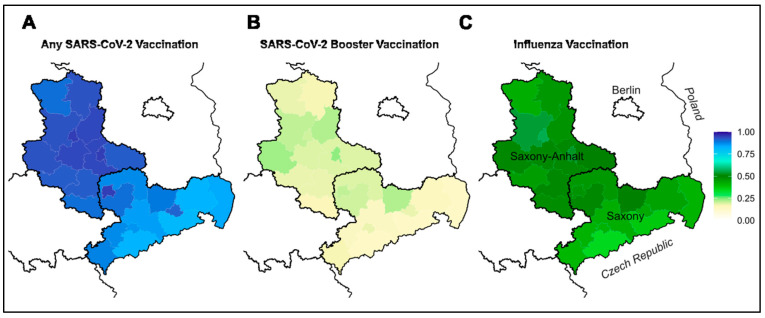
Regional vaccination uptake within Saxony and Saxony-Anhalt, where participants from all administrative district units (Landkreis) are included in the DigiHero study: (**A**) any SARS-CoV-2 Vaccination; (**B**) any SARS-CoV-2 Booster Vaccination; and (**C**) Influenza Vaccination (displayed as proportion vaccinated).

**Figure 4 vaccines-11-01640-f004:**
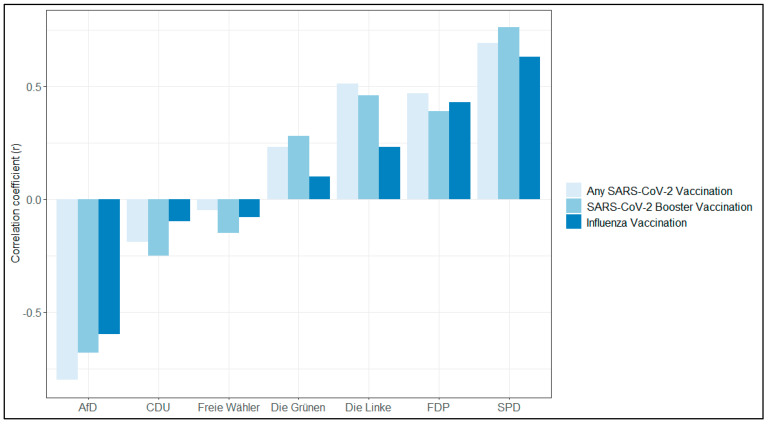
Correlations between regional uptake of vaccines and political support for the seven strongest parties in Saxony and Saxony-Anhalt (AfD–Alternative for Germany; CDU–Christian Democratic Union of Germany; Freie Wähler–Free Voters; Die Grünen–The Green Party; Die Linke–The Left (party); FDP–Democrats; SPD–Social Democratic Party of Germany).

**Figure 5 vaccines-11-01640-f005:**
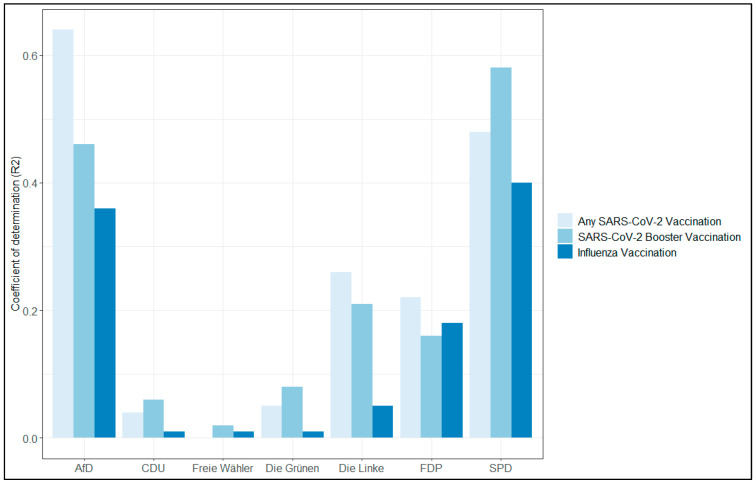
Explained variance in regional uptake of vaccines by political support for the seven strongest parties in Saxony and Saxony-Anhalt (AfD–Alternative for Germany; CDU–Christian Democratic Union of Germany; Freie Wähler–Free Voters; Die Grünen–The Green Party; Die Linke–The Left (party); FDP–Democrats; SPD–Social Democratic Party of Germany).

**Table 1 vaccines-11-01640-t001:** Sociodemographic characteristics of responders and non-responders.

Characteristic	Value	Responders N (*n* = 37,078)	Responders % (95% CI *)	Non-Responders N (*n* = 33,476)	Non-Responders % (95% CI *)
Sex	Male	14,216	38.3 (37.8–38.8)	14,984	44.8 (44.2–45.3)
	Female	22,583	60.9 (60.4–61.4)	17,308	51.7 (51.2–52.2)
	Diverse	29	0.1 (0.1–0.1)	77	0.2 (0.2–0.3)
	NA	250	0.7 (0.6–0.8)	1107	3.3 (3.1–3.5)
Age group	18–29	2633	7.1 (6.8–7.4)	5166	15.4 (15.0–15.8)
	30–39	5079	13.7 (13.4–14.1)	6193	18.5 (18.1–18.9)
	40–49	6065	16.4 (15.9–16.7)	5499	16.4 (16.0–16.8)
	50–59	9241	24.9 (24.5–25.4)	6264	18.7 (18.3–19.1)
	60–69	8831	23.8 (23.3–24.3)	5327	15.9 (15.5–16.3)
	70–79	4023	10.9 (10.5–11.2)	2990	8.9 (8.6–9.2)
	80+	890	2.4 (2.2–2.6)	854	2.6 (2.4–2.7)
	NA	316	0.9 (0.8–0.9)	1183	3.5 (3.3–3.7)
Education ^a^	Low	840	2.3 (2.1–2.4)	2177	6.5 (6.2–6.8)
	Medium	10,714	28.9 (28.4–29.4)	10,223	30.5 (30.0–31.0)
	High	23,319	62.9 (62.4–63.4)	18,396	54.9 (54.4–55.5)
	NA	2205	5.9 (5.7–6.2)	2680	8.0 (7.7–8.3)
Net household equivalent income ^b^	Below 1250 €	3607	9.8 (9.4–10.0)	4958	14.8 (14.4–15.2)
	1250- <1750 €	7966	21.5 (21.1–21.9)	7229	21.6 (21.2–22.0)
	1750- <2250 €	5061	13.6 (13.3–14.0)	4405	13.2 (12.8–13.5)
	2250- <3000 €	9170	24.7 (24.3–25.2)	7115	21.2 (20.8–21.7)
	3000- <4000 €	7568	20.4 (20.0–20.1)	5089	15.2 (14.8–15.6)
	4000- <5000 €	293	0.8 (0.7–0.9)	200	0.6 (0.5–0.7)
	>5000 €	223	0.6 (0.5–0.7)	154	0.5 (0.4–0.5)
	NA	3190	8.6 (8.3–8.9)	4326	12.9 (12.6–13.3)
Born in Germany	Yes	35,678	96.2 (96.0–96.4)	30,866	92.2 (91.9–92.4)
	No	1165	3.1 (2.9–3.3)	1507	4.5 (4.3–4.7)
	NA	235	0.6 (0.6–0.7)	1103	3.3 (3.1–3.5)
Living in a city > 100,000	Yes	13,210	35.6 (35.1–36.1)	13,084	39.0 (38.6–39.6)
	No	23,322	62.9 (62.4–63.4)	19,731	58.9 (58.4–59.5)
	NA	546	1.5 (1.4–1.6)	661	1.9 (1.8–2.1)
Federal state	Saxony-Anhalt	9225	24.9 (24.4–25.3)	10,201	30.5 (29.9–30.9)
	Saxony	7633	20.6 (20.2–21.0)	7143	21.3 (20.9–21.8)
	Baden-Württemberg (only city of Stuttgart)	459	1.2 (1.1–1.4)	518	1.6 (1.4–1.7)
	Bavaria	3329	9.0 (8.7–9.3)	3675	10.9 (10.6–11.3)
	Berlin	479	1.3 (1.2–1.4)	456	1.4 (1.2–1.5)
	Brandenburg	3272	8.8 (8.5–9.1)	2425	7.2 (6.9–7.5)
	Hamburg	509	1.4 (1.3–1.5)	501	1.5 (1.4–1.6)
	Rhineland-Palatinate	4352	11.7 (11.4–12.1)	3676	10.9 (10.6–11.3)
	Mecklenburg-Western Pomerania	2071	5.6 (5.4–5.8)	1456	4.3 (4.1–4.6)
	Lower Saxony	1816	4.9 (4.7–5.1)	999	2.9 (2.8–3.2)
	Saarland	1134	3.1 (2.9–3.2)	560	1.7 (1.5–1.8)
	Schleswig-Holstein	2162	5.8 (5.6–6.1)	1136	3.4 (3.2–3.6)
	Other	91	0.2 (0.2–0.3)	69	0.2 (0.2–0.3)
	NA	546	1.5 (1.4–1.6)	661	1.9 (1.8–2.1)

* CI–confidence interval. ^a^ We categorized education based on the International Standard Classification of Education (ISCED-97) [15]. ^b^ In the baseline questionnaire, we inquired about net household income, providing seven income categories in Euros. To determine the net household equivalent income, we divided the mean of each income category by the sum of the household weights. Household weights were calculated using the following criteria: the first adult was assigned a weight of “1”, while all other adults (individuals aged 14 and above) held a weight of “0.5”, and children were assigned a weight of “0.3”. The resulting value was then categorized using the initial seven income categories.

**Table 2 vaccines-11-01640-t002:** Vaccination uptake for any SARS-CoV-2 vaccinations, SARS-CoV-2 booster vaccinations, and influenza vaccinations.

Characteristics	Value	Any SARS-CoV-2 Vaccination % (95% CI)	SARS-CoV-2 Booster Vaccination % (95% CI)	Influenza Vaccination % (95% CI)
Sex	Male	93.5 (93.0–93.9)	31.8 (31.1–32.6)	47.8 (46.9–48.6)
	Female	92.5 (92.1–92.8)	21.6 (21.1–22.2)	41.8 (41.2–42.5)
	Diverse	93.1 (75.8–98.8)	20.7 (8.7–40.3)	27.6 (13.4–47.5)
	NA	90.4 (85.9–93.6)	26.8 (21.5–32.8)	41.6 (35.5–47.9)
Age group	18–29	94.7 (93.7–95.5)	8.1 (7.1–9.2)	19.6 (18.1–21.2)
	30–39	90.5 (89.7–91.2)	10.1 (9.3–11.0)	24.9 (23.7–26.1)
	40–49	89.6 (88.8–90.3)	12.1 (11.4–13.0)	32.9 (31.8–34.2)
	50–59	91.8 (91.2–92.3)	16.5 (15.8–17.3)	40.9 (39.9–41.9)
	60–69	94.7 (94.2–95.2)	39.7 (38.7–40.7)	58.4 (57.3–59.4)
	70–79	96.8 (96.2–97.3)	57.3 (55.7–58.8)	69.7 (68.2–71.1)
	80+	98.5 (97.4–99.2)	68.7 (65.5–71.7)	77.6 (74.7–80.3)
	NA	87.9 (83.7–91.2)	21.2 (16.9–26.2)	42.7 (37.3–48.4)
Education	Low	96.1 (94.5–97.2)	9.4 (7.6–11.6)	21.9 (19.1–24.9)
	Medium	90.7 (90.1–91.2)	21.8 (21.0–22.6)	38.4 (37.5–39.4)
	High	93.9 (93.7–94.3)	27.8 (27.2–28.3)	47.8 (47.2–48.5)
	NA	89.7 (88.3–90.9)	26.7 (24.8–28.6)	
Net equivalent income	Below 1250 €	88.9 (87.7–89.9)	17.4 (16.2–18.7)	33.1 (31.6–34.7)
	1250- <1750 €	91.5 (90.8–92.1)	26.3 (25.3–27.3)	45.0 (43.9–46.1)
	1750- <2250 €	91.7 (90.9–92.4)	16.6 (15.6–17.7)	36.6 (35.2–37.9)
	2250- <3000 €	94.2 (93.7–94.7)	26.5 (25.6–27.5)	46.4 (45.4–47.4)
	3000- <4000 €	95.9 (95.5–96.4)	33.6 (32.5–34.7)	51.0 (49.9–52.2)
	4000- <5000 €	95.9 (92.8–97.8)	30.0 (24.9–35.7)	50.5 (44.6–56.4)
	>5000 €	95.9 (92.2–98.0)	33.2 (27.1–39.8)	50.7 (43.9–57.4)
	NA	90.7 (89.6–91.6)	24.4 (22.9–25.9)	41.9 (40.2–43.7)
Born in Germany	Yes	92.8 (92.6–93.1)	25.3 (24.9–25.8)	44.1 (43.6–44.6)
	No	94.2 (92.6–95.4)	32.9 (30.2–35.7)	43.2 (40.3–46.1)
	NA	86.4 (81.2–90.4)	23.4 (18.3–29.4)	43.4 (37.0–50.0)
Living in a city > 100,000	Yes	95.4 (95.0–95.8)	27.5 (26.7–28.3)	46.4 (45.5–47.2)
	No	91.4 (91.0–91.8)	24.5 (23.9–25.1)	42.8 (42.2–43.5)
	NA	91.8 (89.0–93.9)	22.9 (19.5–26.7)	43.8 (39.6–48.1)
Federal state	Saxony-Anhalt	93.8 (93.2–94.3)	20.1 (19.3–20.9)	50.3 (49.3–51.4)
	Saxony	86.2 (85.3–86.9)	16.3 (15.5–17.2)	41.9 (40.8–43.0)
	Baden-Württemberg (only Stuttgart)	96.1 (93.8–97.6)	30.0 (25.9–34.5)	38.8 (34.3–43.4)
	Bavaria	92.8 (91.9–93.7)	24.1 (22.7–25.6)	32.5 (30.9–34.1)
	Berlin	97.1 (95.0–98.3)	35.7 (31.4–40.2)	47.8 (43.2–52.4)
	Brandenburg	92.8 (91.9–93.7)	24.9 (23.5–26.5)	47.4 (45.6–49.1)
	Hamburg	97.2 (95.3–98.4)	42.4 (38.1–46.9)	52.1 (47.6–56.4)
	Rhineland-Palatinate	96.4 (95.8–96.9)	35.4 (33.9–36.9)	40.2 (38.8–41.7)
	Mecklenburg-Western Pomerania	92.6 (91.4–93.7)	27.6 (25.7–29.6)	45.3 (43.2–47.5)
	Lower Saxony	96.9 (95.9–97.5)	36.3 (34.1–38.6)	43.3 (41.0–45.7)
	Saarland	97.0 (95.8–97.9)	32.4 (29.7–35.3)	39.9 (37.1–42.9)
	Schleswig-Holstein	97.2 (96.4–97.9)	43.8 (41.7–45.9)	46.7 (44.6–48.8)
	Other	97.8 (91.5–99.6)	25.3 (17.0–35.7)	26.4 (17.9–36.8)
	NA	91.8 (89.0–93.9)	22.9 (19.5–26.7)	43.8 (39.6–48.1)

## Data Availability

The anonymized data reported in this study can be obtained from the corresponding author upon request. The dataset includes individual data and an additional data dictionary will be provided. The beginning of data availability starts with the date of publication and the authors will support any requests in the three following years. Data requests should include a proposal for the planned analyses. Decisions will be made according to data use by the access committee of the DigiHero study and data transfer will require a signed data access agreement.

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
