# Peer review of "Regional Differences in Uptake of Vaccination against COVID-19 and Influenza in Germany: Results from the DigiHero Cohort"

_vaccines, 2023, doi:10.3390/vaccines11111640_

Round 1
Reviewer 1 Report
This is interesting and challenging ecologic study attempting to explore how much of the variation in vaccine coverage could be explained by differences in sociodemographic characteristics including age, sex, and extending this information to education and income. Additionally, they correlated their findings with state election results and voting for seven different political parties. Although authors outlined contextual background, they did not formulate clear hypothesis or aim of the study. To explore possible factors contributing to vaccine hesitancy they analyzed individual data within a cohort of 37,078 participants from 13 German federal states in the digital health cohort study (DigiHero). Rate of vaccination and indirectly vaccine hesitancy was analyzed for 3 different vaccinations: in three separate categories: 1). any SARS-CoV-2 Vaccination; 2). any SARS-CoV-2 Booster Vaccination; and 3). Influenza Vaccination.
There was significant number of data analyzed showing regional variation in vaccine uptake with some correlation for sociodemographic variables such as age, income and living in big cities. However, these findings had limited explanatory power regarding regional differences in vaccine uptake regarding age and sex as they remained largely unchanged after analyzing income and education status. Aiming to offer plausible explanation to these regional differences authors proceeded to corelate individual data with percentage of support for the political parties in the state elections. This ecologic analysis approach showed a rather strong correlation with the regional proportion of first votes in the last local elections, with three parties with a positive correlation, two with no correlation and one with a strong negative correlation in the two federal states studied. Authors concluded could explain considerable fraction of variation in vaccination. The latter finding while plausible, can be misleading as election data from the federal states are being studies with individual data and as such in ecologic analysis interferences cannot be made about individual study participants.
Additional queries:
Figure 2 clarity may benefit with replacing Model 1, Model 2… with specific definition of the 1. State 2. Age respectively.
Author Response
Thank you very much for taking the time to review this manuscript. Please find the detailed responses below and the corresponding revisions/corrections highlighted in the re-submitted files.
Please note that our responses are interspaced and in bold.
General comments: This is interesting and challenging ecologic study attempting to explore how much of the variation in vaccine coverage could be explained by differences in sociodemographic characteristics including age, sex, and extending this information to education and income. Additionally, they correlated their findings with state election results and voting for seven different political parties.
Specific comments:
- Although authors outlined contextual background, they did not formulate clear hypothesis or aim of the study. To explore possible factors contributing to vaccine hesitancy they analyzed individual data within a cohort of 37,078 participants from 13 German federal states in the digital health cohort study (DigiHero). Rate of vaccination and indirectly vaccine hesitancy was analyzed for 3 different vaccinations: in three separate categories: 1). any SARS-CoV-2 Vaccination; 2). any SARS-CoV-2 Booster Vaccination; and 3). Influenza Vaccination.
We thank the reviewer for this comment. We have made changes to the aim of the study as suggested by the reviewer. This can be found in the Introduction section (lines 104-109):
“Given this contextual background, our objective was to investigate how much of the regional differences across federal states or within regions in Germany could be explained by differences in sociodemographic factors for COVID-19, COVID-19 booster, and influenza vaccinations. Furthermore, we studied whether regional differences in vaccine uptake were associated with support for the specific political parties at the regional level.” - There was significant number of data analyzed showing regional variation in vaccine uptake with some correlation for sociodemographic variables such as age, income and living in big cities. However, these findings had limited explanatory power regarding regional differences in vaccine uptake regarding age and sex as they remained largely unchanged after analyzing income and education status. Aiming to offer plausible explanation to these regional differences authors proceeded to correlate individual data with percentage of support for the political parties in the state elections. This ecologic analysis approach showed a rather strong correlation with the regional proportion of first votes in the last local elections, with three parties with a positive correlation, two with no correlation and one with a strong negative correlation in the two federal states studied. Authors concluded could explain considerable fraction of variation in vaccination. The latter finding while plausible, can be misleading as election data from the federal states are being studies with individual data and as such in ecologic analysis interferences cannot be made about individual study participants.
We thank the reviewer for this comment. We acknowledge that correlating individual data with the percentage of support for political parties in state elections can be misleading, as it does not allow us to draw conclusions about the political preferences of individual study participants. We emphasized that our study employs an ecological approach. This can be found in the Materials and Methods section, Study Design and Questionnaires subsection (lines 140-142):
“We then retrieved results from the state elections in Saxony in 2019 and Saxony-Anhalt in 2021 at the administrative district (Landkreis) level from official election sources [18, 19].”
We also made changes in the Materials and Methods section, Statistical Analysis subsection (lines 155-158):
“For the ecological analysis, we examined the regional variations in vaccination uptake within the federal states of Saxony and Saxony-Anhalt, both of which had participants from all administrative district units (Landkreis) in the DigiHero study. We then correlated these variations with the percentage of support for political parties.”
Additionally, we recognize the drawbacks of this ecological approach and have added changes to address this. This can be found in the Discussion section (lines 321-327):
“In our analysis, we compared vaccination proportions among study participants with external voting data. It is important to note that this approach may be susceptible to ecological fallacy. This was a simple correlation intended to provide an overall perspective rather than draw specific conclusions about the individual study participants’ political preferences. When correlating individual data with the percentage of support for political parties in state elections, we cannot infer the political preferences of individual participants.” - Figure 2 clarity may benefit with replacing Model 1, Model 2… with specific definition of the 1. State 2. Age respectively.
We thank the reviewer for this comment. We have made adjustments to the figure description. This can be found in the Results section (lines 205-217):
“Figure 2. Effects of adjusting for sociodemographic variables on regional variation in vaccine uptake across different federal states in Germany for: (A) any SARS-CoV-2 Vaccination; (B) any SARS-CoV-2 Booster Vaccination; and (C) Influenza Vaccination. The odds ratios are presented, from logistic regression using Saxony-Anhalt as the reference category. The models progress from Model 1 to 6, each adjusting for an increasing number of variables: (i) Model 1 adjusts for only the federal state; (ii) Model 2 adjusts for the federal state and age; (iii) Model 3 adjusts for the federal state, age, and net equivalent household income; (iv) Model 4 adjusts for the federal state, age, net equivalent household income, and living in city (>100,000 inhabitants); (v) Model 5 adjusts for the federal state, age, net equivalent household income, living in city (>100,000 inhabitants), and education; (vi) Model 6 adjusts for the federal state, age, net equivalent household income, living in city (>100,000 inhabitants), education, and sex. Additionally, the age variable was adjusted with splines.”
Reviewer 2 Report
The problem of vaccination hesitancy is a sociological problem, with strong connection with the conspiracy theories. This is an issue of fundamental interest going even beyond the important problem of vaccination hesitancy favoring the spread of Covid-19 epidemics despite the scientifically funded efficiency of vaccination.
This manuscript addresses the problem of vaccination hesitancy in Germany, with the main conclusion that in addition to significant sociodemographic factors, political stances play a major role. The authors noticed correlation between support of political parties and regional vaccination uptake.
This observation makes this manuscript very important not only for vaccination hesitancy, but more in general for the political influence on biomedical issues concerning health, involving social psychology and medical anthropology. I warmly recommend this paper for publication.
Author Response
Thank you very much for taking the time to review this manuscript. Please find the detailed responses below and the corresponding revisions/corrections highlighted in the re-submitted files.
Please note that our responses are interspaced and in bold.
General comments: The problem of vaccination hesitancy is a sociological problem, with strong connection with the conspiracy theories. This is an issue of fundamental interest going even beyond the important problem of vaccination hesitancy favoring the spread of Covid-19 epidemics despite the scientifically funded efficiency of vaccination.
Specific comments:
- This manuscript addresses the problem of vaccination hesitancy in Germany, with the main conclusion that in addition to significant sociodemographic factors, political stances play a major role. The authors noticed correlation between support of political parties and regional vaccination uptake.
We thank the reviewer for this comment. While our study found that sociodemographic factors had limited explanatory power regarding regional differences in vaccine uptake, we concluded that political preferences might offer a plausible explanation. However, because individual political preferences were not included in the questionnaire, we relied on external voting data. - This observation makes this manuscript very important not only for vaccination hesitancy, but more in general for the political influence on biomedical issues concerning health, involving social psychology and medical anthropology. I warmly recommend this paper for publication.
We thank the reviewer for this comment and for their acknowledgement of the significance of political influence on biomedical issues.
Reviewer 3 Report
Thank you for your article, which investigates the regional disparities in vaccination uptake during the COVID-19 pandemic in Germany. The paper explores the impact of sociodemographic variables and political dynamics on vaccination behavior.
The introduction could provide more details, in order not to confuse hesitancy and risk-benefits. Since later the political dynamics will be explored, it would be useful to provide a short overview on the political landscape in Germany at that time.
The study design is well described. I also appreciate that the characteristics of non-responders are analysed.
For the final version, a better layout for the tables, text and figures would be suitable (e.g. not over 2 pages if possible, or with new headings)
The discussion puts the covid-vaccine hesitancy issue in a broader vaccine context, and it is interesting to note the apparent paradox in the former Eastern German states.
The limitations are well explained.
Some suggestions:
Introduction:
line 71-74: the fact that age plays a fundamental role for a second booster has more to do with recommendations and risk-benefit, than with vaccine hesitancy (other than for basic immunization). I suggest evoking this fact, in order to differentiate difference in vaccine hesitancy with higher rates due to higher risk.
Discussion:
line 266-268: this is not so clear and must not be stated without sound reference! (I'd suggest leaving it out)
In summary, this is a good article pointing at the importance of beliefs and values in vaccine uptake.
Author Response
Thank you very much for taking the time to review this manuscript. Please find the detailed responses below and the corresponding revisions/corrections highlighted in the re-submitted files.
Please note that our responses are interspaced and in bold.
General comments: The paper explores the impact of sociodemographic variables and political dynamics on vaccination behavior. In summary, this is a good article pointing at the importance of beliefs and values in vaccine uptake.
Specific comments:
- The introduction could provide more details, in order not to confuse hesitancy and risk-benefits. Since later the political dynamics will be explored, it would be useful to provide a short overview on the political landscape in Germany at that time.
We thank the reviewer for this comment. We added the difference between vaccine hesitancy and risk-benefits assessment of vaccinations but we did not go into detail since this paper focuses on vaccine hesitancy. This can be found in the Introduction section (lines 61-68):
“At the same time, it is crucial to differentiate vaccine hesitancy from the risk-benefit assessment of vaccines. Vaccine hesitancy focuses on people’s attitudes and perceptions regarding vaccination [3], while the risk-benefit assessment involves a systematic comparison between the risks and benefits of vaccination [6]. This is essential for informing public health decision-making [6]. Both aspects, hesitancy and risk-benefit assessment can play a role in vaccine uptake. An additional aspect of judging risk-benefit is the effect of vaccinations on the spread of infection, thus going beyond individual perspective.”
While we did not provide in-depth details of vaccine hesitancy in the Introduction section, we have addressed this topic in detail within the Discussion section (lines 264-276):
“In fact, it appears that beliefs related to vaccination are more profound and related to personal preferences. An in-depth qualitative survey involving 33 German participants identified four primary themes that influenced the decision to receive a COVID-19 vaccine: (i) assessment of the benefits and risks of being vaccinated against COVID-19; (ii) influence of existing social and political conditions; (iii) emotional responses to the pandemic and its’ social and political impact; and (iv) trust and confidence in health authorities and the vaccines themselves [20]. Recurrent themes across multiple studies in both Germany and Europe as a whole included reasons such as low levels of trust in healthcare authorities and COVID-19 vaccines, concerns about potential side effects, and alignment with right-wing political stances [21-26]. Thus, it appears that there are fundamental concerns about vaccination that are also intertwined with political ideologies. It is not clear in which direction the influence between politics and vaccination concerns occurs.”
Additionally, also provided a short overview on the political landscape in Germany at that time. This can be found in the Introduction section (lines 74-84):
“In Germany, there are notable disparities between former Eastern and Western federal states with respect to vaccinations. Vaccine hesitancy towards COVID-19 vac-cines had a particularly noticeable presence in the former Eastern German states [2]. Eastern German states tend to have stronger support for right-wing parties, while Western German states lean more toward centrist political parties [2, 10]. AfD (Alter-native for Germany), the right-wing German political party, along with other protest parties that emerged during the COVID-19 pandemic, have made public their re-sistance to government measures and the associated vaccines [2]. As of April 2023, the overall vaccination coverage in Germany was 77.9%, with people of all ages receiving at least one dose [11]. Furthermore, 76.4% of the total population received basic im-munization, 62.6% received one or more booster doses, and 15.2% received at least two booster doses [11].” - For the final version, a better layout for the tables, text and figures would be suitable (e.g. not over 2 pages if possible, or with new headings)
We thank the reviewer for this comment. We have made adjustments to the table layout, which include inserting additional lines between the columns and adding shading to the headers. To accommodate the content, the tables could not be condensed to a single page so we have introduced separate headers when the tables continue onto the next page. This can be found in the Results section (pages 5-8). Furthermore, we have included borders around the figures to enhance readability. Additionally, to ensure clarity, we have positioned the figure descriptions below each figure, making it clear which figure number and description correspond to each figure. This can be found in the Results section (pages 9-12). - The limitations are well explained.
We thank the reviewer for this comment. We have added additional limitations on the ecological approach of our study. This can be found in the Discussion section (lines 303-312):
“Our study had several limitations. Firstly, our data included only information on vaccination uptake, we did not assess vaccine hesitancy. Therefore, we can only indirectly infer on vaccination hesitancy from regional variation, after controlling for regional variation in sociodemographic factors. Furthermore, at individual level, vaccine hesitancy can be intertwined with an individual risk-benefit assessment. An even more complex issue results from transmissibility of infections and the risk-benefit assessment can include considerations beyond one’s own health. Furthermore, at a more technical level, we considered only the number of vaccinations for COVID-19, while particularly in the early stages of the pandemic, infections were considered to provide protection for a certain period. This approach might have led to an underestimation of willingness for vaccination.” - Introduction: line 71-74: the fact that age plays a fundamental role for a second booster has more to do with recommendations and risk-benefit, than with vaccine hesitancy (other than for basic immunization). I suggest evoking this fact, in order to differentiate difference in vaccine hesitancy with higher rates due to higher risk.
We thank the reviewer for this comment. We have removed the following sentences from the Introduction section: “Moreover, age played a fundamental role in the uptake of vaccinations. While individuals above the age of 60 demonstrated higher rates of acquiring basic immunization (90.1%) and obtaining at least two booster vaccinations (39%), those in the 18 to 59 years age group exhibited lower rates of 83% for basic immunization and only 6.8% for two booster doses [9].” - Discussion: line 266-268: this is not so clear and must not be stated without sound reference! (I'd suggest leaving it out)
We thank the reviewer for this comment. We have removed the following statement from the Discussion section: “but the protection against a severe course of infection offered by a preceding infection is considered to last shorter than by vaccination. This situation might lead to a substantial burden of severe infections.” The overall sentence in this paragraph has also been changed for clarity. This can be found in the Discussion section (lines 293-297):
“The low uptake of SARS-CoV-2 booster vaccination in the autumn/winter 2022/23 suggests that a significant portion of the German population will have received their last vaccination more than 12 months ago by the autumn/winter of 2023/24. This is concerning because, during the winter season of 2022/23, a significant number of people continued to become infected with SARS-CoV-2 [28].”
Reviewer 4 Report
This is an interesting questionnaire-based study on vaccine hesitancy in Germany with particular reference to the Covid-19 pandemic. The study used an impressively large cohort of participants (37,078). Rates of COVID-19 primary vaccination and booster vaccination were compared with influenza vaccination and assessed the contributing factors including the influence of sociodemographic variables on primary COVID-19, COVID-19 booster, and influenza vaccinations within a cohort of participants from 13 German federal states in the digital health cohort study, DigiHero. The study found substantial correlations between regional support of specific parties during the last local elections and the vaccination uptake according to representational zones. There were other interesting findings related to sociodemographics.
I am happy with the overall content of the paper, however, there are a couple of minor issues, for instance the key attached to Tables 4 and 5 contains the following
“(SPD- Social Democratic Party of Germany; CDU- Christian Democratic Union of Germany; Freie Wähler- Free Voters; Die Grünen – The Green Party; Die Linke – The Left (Party); FDP – Democrats; SPD- Social Democratic Party of Germany).”
As can be seen, the SPD occurs twice and the AfD is omitted.
Perhaps the authors could define "LandKreis"
Author Response
Thank you very much for taking the time to review this manuscript. Please find the detailed responses below and the corresponding revisions/corrections highlighted in the re-submitted files.
Please note that our responses are interspaced and in bold.
General comments: This is an interesting questionnaire-based study on vaccine hesitancy in Germany with particular reference to the Covid-19 pandemic. The study used an impressively large cohort of participants (37,078). Rates of COVID-19 primary vaccination and booster vaccination were compared with influenza vaccination and assessed the contributing factors including the influence of sociodemographic variables on primary COVID-19, COVID-19 booster, and influenza vaccinations within a cohort of participants from 13 German federal states in the digital health cohort study, DigiHero. The study found substantial correlations between regional support of specific parties during the last local elections and the vaccination uptake according to representational zones. There were other interesting findings related to sociodemographics.
Specific comments:
- I am happy with the overall content of the paper, however, there are a couple of minor issues, for instance the key attached to Tables 4 and 5 contains the following: “(SPD- Social Democratic Party of Germany; CDU- Christian Democratic Union of Germany; Freie Wähler- Free Voters; Die Grünen – The Green Party; Die Linke – The Left (Party); FDP – Democrats; SPD- Social Democratic Party of Germany).” As can be seen, the SPD occurs twice and the AfD is omitted.
We thank the reviewer for this comment. We have corrected the figure descriptions. This can be found below Figure 4 and Figure 5, in the Results section (lines 242-246 & 248-252):
“Figure 4. Correlations between regional uptake of vaccines and political support for the seven strongest parties in Saxony and Saxony-Anhalt (AfD – Alternative for Germany; CDU – Christian Democratic Union of Germany; Freie Wähler – Free Voters; Die Grünen – The Green Party; Die Linke – The Left (party); FDP – Democrats; SPD – Social Democratic Party of Germany).”
“Figure 5. Explained variance in regional uptake of vaccines by political support for the seven strongest parties in Saxony and Saxony-Anhalt (AfD – Alternative for Germany; CDU – Christian Democratic Union of Germany; Freie Wähler – Free Voters; Die Grünen – The Green Party; Die Linke – The Left (party); FDP – Democrats; SPD – Social Democratic Party of Germany).”
Perhaps the authors could define "LandKreis"
We thank the reviewer for this comment. We added that Landkreis is the administrative district level. This can be found in the Materials and Methods section, Study Design and Questionnaires subsection (lines 140-142):
“We then retrieved results from the state elections in Saxony in 2019 and Saxo-ny-Anhalt in 2021 at the administrative district (Landkreis) level from official election sources [18, 19].”
Round 2
Reviewer 1 Report
This is significantly improved version of interesting and highly relevant study. Authors addressed all the queries that made objective of the study clear. Well-designed work with impressive data presentations